# Different Machine Learning Approaches for Implementing Telehealth-Based Cancer Pain Management Strategies

**DOI:** 10.3390/jcm11185484

**Published:** 2022-09-19

**Authors:** Marco Cascella, Sergio Coluccia, Federica Monaco, Daniela Schiavo, Davide Nocerino, Mariacinzia Grizzuti, Maria Cristina Romano, Arturo Cuomo

**Affiliations:** 1Department of Anesthesia and Critical Care, Istituto Nazionale Tumori—IRCCS, Fondazione Pascale, 80100 Naples, Italy; 2Department of Electrical Engineering and Information Technologies—DIETI, University Federico II, 80138 Naples, Italy; 3Epidemiology and Biostatistics Unit, Istituto Nazionale Tumori—IRCCS, Fondazione Pascale, 80100 Naples, Italy

**Keywords:** telemedicine, telehealth, teleconsultations, predictive models, machine learning, cancer pain, random forest, gradient boosting machine, artificial neural network, LASSO–RIDGE algorithm

## Abstract

Background: The most effective strategy for managing cancer pain remotely should be better defined. There is a need to identify those patients who require increased attention and calibrated follow-up programs. Methods: Machine learning (ML) models were developed using the data prospectively obtained from a single-center program of telemedicine-based cancer pain management. These models included random forest (RF), gradient boosting machine (GBM), artificial neural network (ANN), and the LASSO–RIDGE algorithm. Thirteen demographic, social, clinical, and therapeutic variables were adopted to define the conditions that can affect the number of teleconsultations. After ML validation, the risk analysis for more than one remote consultation was assessed in target individuals. Results: The data from 158 patients were collected. In the training set, the accuracy was about 95% and 98% for ANN and RF, respectively. Nevertheless, the best accuracy on the test set was obtained with RF (70%). The ML-based simulations showed that young age (<55 years), lung cancer, and occurrence of breakthrough cancer pain help to predict the number of remote consultations. Elderly patients (>75 years) with bone metastases may require more telemedicine-based clinical evaluations. Conclusion: ML-based analyses may enable clinicians to identify the best model for predicting the need for more remote consultations. It could be useful for calibrating care interventions and resource allocation.

## 1. Introduction

Managing cancer-related pain typically requires a complex and multimodal approach [1]. One of the main challenges concerns the development of a useful pathway for addressing the multiple problems that can occur during the disease course [2,3,4].

Given that a model of care based on face-to-face visits requires an important commitment of resources, innovative strategies must be evaluated. Telemedicine may offer a variety of applications to re-evaluate pathways of care, including cancer pain management [5]. In this context, telemedicine-based strategies can have a paramount economic and organizational impact on healthcare systems [6], enhancing the quality of the care provided. A recent evidence-based analysis demonstrated that eHealth interventions are effective in improving pain management [7]. Although during the COVID-19 pandemic, different telemedicine approaches have been proposed [8], there is a need for establishing pathways that are valid beyond the emergency and routinely applied to clinical practice [9,10].

On the other hand, lacking the literature data from large-scale clinical experiences and precise directives from scientific societies, it is difficult to establish a model that provides for the integration of telemedicine in the treatment process. Consequently, a proper strategy for the management of cancer pain through telemedicine should be fully designed.

The use of predictive models represents an important opportunity in medicine. The benefits of artificial intelligence (AI) and its branches such as machine learning (ML) are intended to enhance patient care, but also involve organizational processes and healthcare systems [11]. In the planning of care pathways, AI represents a valuable helpful resource to improve hospital workflows, identifying the activities that require priority and providing an adequate service to the patient’s needs. Recently, for example, it was demonstrated that AI strategies such as natural language processing models can be a reliable guide to trigger early access for uncontrolled pain and other symptoms in palliative care [12].

In a recent cross-sectional investigation, we proposed a model of care and evaluated adherence to the telemedicine pathway [13]. This “hybrid” model provides for scheduled remote visits, but the patient can require other consultations. Additionally, in-person access is provided for emergencies or for diagnostic or clinical aims. For each patient, the number of telemedicine visits can vary depending on an unspecified number of reasons, and we have noticed that some patients required a greater number of remote consultations. On these premises, the purpose of this study is the development of data-driven predictive models for identifying those patients who may require more remote consultations. In the context of precision medicine for cancer pain management, we implemented ML algorithms to better customize treatment strategies. As pieces of evidence are needed to establish the most appropriate telemedicine pathways, the recognition of those patients who require a greater number of remote visits can stimulate the planning of ad hoc processes for managing multiple care needs and calibrating resource allocation.

## 2. Materials and Methods

### 2.1. Study Population

The study population included adult patients treated for cancer pain at the Istituto Nazionale Tumori, Fondazione Pascale, Italy.

A hybrid model of care was implemented. After the first in-person visit for a complete clinical and instrumental evaluation and for addressing legal and regulatory issues (consent acquisition), data collection, and training, a synchronous real-time video consultation was scheduled according to the clinical need. Further remote controls were programmed or required by the patients. Moreover, face-to-face visits were allowed to carry out minimally invasive procedures, for diagnosis, acute clinical motivations (e.g., drug side effects), or if requested by the patient [13].

The local Medical Ethics Committee approved this study (protocol code 41/20 Oss; date of approval, 26 November 2020), and all patients provided written informed consent. The investigation was conducted in accordance with the Declaration of Helsinki.

### 2.2. Data Collection

For each patient, 13 demographic, clinical, and therapeutic variables were collected to investigate the potential causes that may affect the number of remote consultations (Table 1). All the data were reported on a prospectively filled database and then registered on Zenodo [14]. The duration of the study was considered the time interval between the first and last remote consultations. The death of the patient and the occurrence of in-person visits or hospitalization were assumed as the conditions for the end of the observation period for data acquisition. The lack of further remote consultations for a two-month period was another condition for considering the observation closed.

The univariate analysis was performed to detect the main associations of selected features with the outcome variable (remote consultations: one or more).

### 2.3. Predictive Analysis

#### 2.3.1. Preprocessing and Exploratory Data Analysis

After the loading, normalization, and standardization of the dataset (preparation process or preprocessing), as well as an exploratory data analysis aimed at discovering trends, the variables were selected. The expectation–maximization (EM) algorithm was used for the imputation of the missing data [15]. To facilitate model implementations and the interpretation of results, three age groups were obtained by categorizing the variable “age”: ≤55 years old (called “younger patients”), 56–75 years old, and >75 years old (“older patients”).

#### 2.3.2. Machine Learning Algorithms

Four ML-based algorithms were adopted as follows:LASSO–RIDGE regression (elastic model): This is a generalized linear regression model that penalizes a loss function through regressor resizing (16 in all). Most of them are made small or led to zero if not important to explain the dependent variable. This approach reduces model complexity and prevents the over-fitting phenomena [16];Random forest (RF) algorithm: This algorithm can be used for both regression and classification. It is one of the most popular ML methods, belonging to the specific category of bagging methods. RF works on various overall models (decision trees) to improve the performance of each of them individually. The output is the whole contribution from all of them [17];Gradient boosting machine (GBM) is aimed at optimizing previsions by operating on the previous tree regression or classification error and reducing the error function (boosting method). In this way, the succeeding one can improve the prevision skills let by its preceding tree [18];Single hidden layer artificial neural network (ANN): This strategy can minimize a loss function by acting on some weights which tune connections between two neurons of two adjoining layers [19].

We chose these four ML-based prediction models for implementing different methods for regression or classification, such as bagging and boosting (RF and GMB, for additive regression models), and a strong learning method to compare different numerical approaches (LASSO–RIDGE, a binary regression model); ANN is one of the function algorithms that are largely used for classification (and regression) problems.

#### 2.3.3. Model Processing and Evaluation 

Since a predictive analysis was performed in order to predict which cancer patient should need to have more than one remote consultation, the outcome variable was “the number of remote consultations” as dichotomized. Each classifier was optimized by repeated cross-validation (RCV) methods to focus on the best guess through the K-fold mean error calculus and to determine the hyperparameters that support the best guess and identify its structure. The sample was split into a training set (80% of the total size) to identify the hyperparameters and a test set for testing the models (20% of the total size).

A wide choice of hyperparameters was given to every algorithm to finally evaluate the best performance. In particular, each combination of hyperparameters was inserted as input for the algorithm. An 8-fold 5-repeated cross-validation method was adopted to find the best one, so the dataset was divided into 8 parts (20 individuals for any time), and the training and test parts were performed for each combination and for 5 times; the misclassification error rate was calculated upon 5 attempts (for a more precise managing of results). For each algorithm, the following features were applied:GBM: The number of sequential trees from 20 to 100 by 10, tree depth from 2 to 5 shrinkage parameter (regularizing the error function) from 0.01 to 0.1 by 0.01, and a minimum observation-in-a-leaf from 10 to 20 for a total of 3960 were assessed;RF: Only the number of splitting variables was required, which was from 3 to 13;LASSO–RIDGE: Regression alpha and beta were, respectively, given as from 0 to 1 by 0.05 and 0 to 10 by 0.1, for a total of 2121 trials;ANN: This layer was made from 1 to 12 neurons and the decay (a regularization parameter to avoid the over-fitting of weights) ranged from 0.01 to 0.2 by 0.01, for a total of 240 trials.

Comparations were assessed through these models by calculating the accuracy and area under the receiver operating characteristic (ROC) curve (AUC). The AUC represents the sensibility (i.e., TP/(TP+FN)) and 1-specificity (1-FP/(TN+FP)) ratio. The AUC can easily be approximated with the measure of accuracy in the case of equidistribution between the modalities of the employee, but it is also suitable for solving problems of poorly distributed modalities. Each member is comparable to the observed value conditional correct classification rates, respectively. In other words, the AUC is equivalent to the probability that a random positive response is classified with respect to a random negative one. Another adopted goodness-of-prevision statistic parameter was the F1 score:(1)
F1 = 2 * (precision * recall)/(precision + recall)


This measure considers the precision and recall of the test; precision is the number of true positives divided by the number of all the positive results, while recall is the number of true positives divided by the number of all the tests that should have been positive (i.e., true positives plus false negatives). The values of F1 scores range from 0 to 1.

Finally, Mathew’s correlation coefficients (MCC) were calculated (from the confusion matrix) for each model, for obtaining a broad view of their predictive power and robustness (TP, true positive; TN, true negative; FP, false positive; FN, false negative):(2)$$\mathrm{MCC}=\frac{\mathrm{TP}*\mathrm{TN}-\mathrm{FP}*\mathrm{FN}}{\surd \left(\mathrm{TP}+\mathrm{FP}\right)\left(\mathrm{TP}+\mathrm{FN}\right)\left(\mathrm{TN}+\mathrm{FP}\right)\left(\mathrm{TN}+\mathrm{FN}\right)}$$

#### 2.3.4. Risk Analysis

Based on the ML processes, the risk analysis for an increased number (>1) of remote consultations was assessed. We used an odds-ratio-like analysis that we indicated as the simulated odds ratios (SORs). Simulations were assessed in order to evaluate the risk of more consultations in target individuals. In particular, approximately 500 simulations were performed 150 times for creating a classification rate for the cases (target individuals) and control individuals. Subsequently, we calculated the odds ratio as the ratio of the effective odds for each individual typology and 95% credibility intervals (95% CIs) as the effective 2.5 and 97.5 percentiles for the SOR samples. Although wide possibilities were possible, the following four standard clinical conditions (targets) were established:Condition 1: Young patients (≤55 years old) with bone metastases and rapid-acting oral and nasal transmucosal fentanyl formulation (ROO) use (morphine-equivalent dose, MED > 60 mg) for breakthrough cancer pain (BTcP);Condition 2: Older cancer patients (>75 years old), with and without bone metastases;Condition 3:Male and female young patients (≤55 years old) with bone metastases;Condition 4: Younger (≤55 years old) vs. older (>75 years old) patients with bone metastases with gender differences.

### 2.4. Algorithmic Toolkit

The data were analyzed using the *R* software version 4.1.3 (R Core Teams, R Foundation for Statistical Computing, Vienna, Austria). The toolkit included the mice package [20] for the imputation of the missing data. Caret was the main suite used for the implementation (creation, training) and evaluation (testing) of the classifiers [21]. Moreover, purr, pROC, and pRROC [22] were adopted for the construction and visualization of the ROC curves. The graphics packages included ggplot, ggpubr, and cowplot.

## 3. Results

A total of 267 patients were evaluated for cancer pain management through remote consultations between March 2021 and February 2022. Of these patients, 109 were excluded for not being available or having incomplete data; finally, the data from 158 patients were used for the descriptive and predictive analyses (Figure 1).

### 3.1. Descriptive Analysis

The median age was 63 years old. Fifty-one percent were female. Just over half of the patients (53%) had more than one visit. The average number of visits was 2.27, with a standard deviation of 2.05 (Table 2).

The reasons for interruption of the telemedicine pathway (dropouts) were the patient’s death (*n* = 63, 39.9%), the need for an invasive procedure (*n* = 15, 9.5%) or an in-person clinical assessment (*n* = 14, 8.9%). Six patients (3.8%) requested an in-person visit. Unplanned hospital admissions occurred in seven patients (4.4%). About a third of the patients (*n* = 53, 33.5%) were not evaluated (in person or remotely) for at least two months. These patients were contacted (email and telephone), and about half (*n* = 28) did not provide an answer; the remainder (*n* = 25) said they did not need further visits for cancer pain (Figure 2).

The univariate analysis was performed for evaluating the differences between the cohort of patients who underwent one remote consultation and those who received more telemedicine evaluations (Table 3).

### 3.2. Predictive Analysis

Table 4 summarizes the results of the implemented ML methods. In our analyses, the accuracy, that is the proportion of the well-ranked parameters, relative to the training set, reached almost 100% for the RF and ANN algorithms. Nevertheless, the accuracy of the ANN on the test set was reduced by almost 50 percentage points. By contrast, RF showed an acceptable classification level (70% accuracy in the test) (*p* = 0.05) with an F1 score of 0.71.

The overall performance of a classifier, summarized over all the possible thresholds, is given by the area under the ROC curve (AUC). An ideal ROC curve will hug the top left corner: the larger the area under the curve, the better the classifier. Reducing the false-positive rate (FPR) and, at the same time, increasing the true-negative rate (TNR) is like finding a trade-off cut point between the error rates. A classifier that performs worse than a random classification has an AUC statistic of 0.5. Thus, an AUC value closer to 1 indicates a more adequate classification and a lower level of error: Its value is theoretically almost 1 as it is built. The AUC performances of the considered classifiers are reported in Figure 3.

The confusion matrix for the two best models (RF and ANN) during the training phase is shown in Table 5.

### 3.3. Risk Analysis

The model with the best performance (i.e., RF) was implemented for assessing the risk analysis in different scenarios.

Condition 1: We calculated the risk of having repeated remote consultations for different cancer types. The other features were kept as randomly chosen. For those with lung neoplasm, there was a probability of 93.4% (92.6%, 94.2%) to receive multiple consultations and a higher risk (+172.2%, 95% CI = +70%, +301.1%) than cancer patients with no bone metastases; for those patients with colorectal neoplasm, the percentage was 88.8% (88%, 89.7%), and this risk was +92.2% (95%CI = +40.4%, +156.4%). For those affected by other cancers, the percentage was 71.9% (70.6%, 73.3%), and the risk was +55.5% (95%CI = +19.2%, +106.9%). For breast neoplasm with bone metastases, 90.1% (89.2%, 91.2%) of the patients were predicted to have multiple consultations, and +90.6% (95%CI = +32.5%, +158%) was their predicted risk for multiple consultations, compared with those without bone metastasis (Figure 4).

Condition 2: The same analysis was performed for older cancer patients (>75 years old), with and without bone metastases. For those patients affected by lung neoplasm, the risk for multiple remote consultations was 4.4 times (+335.5%, 95%CI = +209%, +529.6%) more than those with no bone metastasis, 2.9 times (+189.4%, 95%CI = +117.6%, +276.9%) for colorectal neoplasm, and 4.6 times (+357.9%, 95%CI = +252.6%, +495.5%) for other types of cancer. The expected probabilities were 88.7% (87.6%, 89.6%) for lung cancer, 82.9% (81.6%, 84%) for colorectal cancer, and 69.4% (68.0%, 70.8%) for other cancers. For those with breast neoplasm with bone metastasis, the simulated percentage of multiple remote consultations was 88.7% (87.8%, 89.6%) with a higher risk (+82.3%, 95%CI = +31.6%, +146.8%) of multiple remote evaluations than those without bone metastases (Figure 5). 

Condition 3: The model demonstrated that male cancer patients can have an 11 times higher risk to receive multiple remote consultations than female cancer patients. The SORs were 11.3 (95%CI = 4.6, 24.1) for lung neoplasm and 11.1 (95%CI = 5.9, 20.6) for colorectal neoplasm. No statistical significance was found for other cancers (SOR = 0.97, 95%CI = 0.76, 1.23) (Figure 6). 

Condition 4: An overall higher risk of having multiple telemedicine visits was found for young cancer patients than for male cancer patients, with SORs of +88.9% (95%CI = +16%, +182%) for lung cancer; +70.1% (95%CI = +24.1%, +143.3%) for colorectal cancer; and +16.5% (95%CI = −0.9%, +51.9%) for other cancers. Compared with older patients, for young female breast cancer patients, no significant risk was found (+19.5%, 95%CI = −20%, +65.2%) (Figure 7).

## 4. Discussion

In the setting of patients suffering from cancer pain, the applications of telemedicine strategies can enhance the effectiveness of clinical management [13] and lead to the optimization of resources [23]. Nevertheless, despite the growing use of telehealth methods, scientific evidence is still scarce to design care pathways. 

Previously, we evaluated patient satisfaction with telemedicine and found high satisfaction rates with the care provided and the platform used. The dropout from the telemedicine pathway was investigated, and we found that approximately 10% of patients leave the telemedicine process due to unplanned clinic or hospital readmission or the need for non-pharmacological treatments [13]. Therefore, in this clinical setting, the development of telemedicine-based programs must consider multiple factors. The proposed model of care provides for a variable number of telemedicine visits by combining scheduled consultations and patient requests. In-person visits can be required to carry out minimally invasive procedures, diagnoses, or for other purposes. Furthermore, access to the hospital is provided for acute clinical conditions. Nevertheless, by following this approach, clinical practice has suggested that the careful planning of controls and the design of a safety pathway is a fundamental preliminary phase for validating our telemedicine-based model of care. The aim is to design a model of care that is generalizable while guaranteeing a patient-centered treatment. 

In the clinical practice through telemedicine, we observed that many cancer patients had just one consultation. However, some individuals required a large number of closely remote visits. Consequently, we decided to evaluate the typology of cancer patients who may require more than one remote consultation. We searched for a more suitable strategy useful for achieving internal and external validation and translating the chosen model into the clinic [24]. For this aim, we adopted different ML models and decided to categorize the number of remote consultations as “one” or “more than one” remote consultation. Furthermore, the prediction of the number of remote consultations for new patients can involve several practical implications, including the design of personalized paths and optimal resource allocation. An increased number of remote consultations for cancer pain management may also reflect on ad hoc public or private healthcare/insurance health programs. For example, in Italy, the Ministry of Health released guidelines for the provision of telemedicine services, stimulating the design of paths for different care needs [25]. 

In ML analyses, preprocessing and exploratory data analysis (EDA) are the key elements of the process and take a large part of the time used for the whole analysis. The variable analysis is a crucial point for the modeling: It is part of the data quality and has a significant influence on the model’s predictive power, robustness, and confidence. During these phases, it emerged that the variable “age” offered useful information for the model construction and understanding. This variable was categorized into three age groups (younger, mean age, and older patients). Consequently, in the univariate analysis, it was found that younger patients underwent more visits (*p* = 0.03). These data were used in the predictive analysis (simulations) for assessing, in target individuals, the risk of having multiple remote consultations. For example, the application of the chosen model (RF) showed that younger patients (≤55 years old) with bone metastases and ROO administration for BTcP treatment have an increased risk for more consultations, especially if affected by lung cancer. These data confirm what we previously highlighted in an analysis focused on the BTcP phenomenon. In a hierarchical classification, the worst phenotype of cancer pain patients was characterized by the presence of BTcP, younger age, and lung cancer [26].

In patients of advanced age (>75 years), the variable “bone metastasis” affected the prediction of the number of visits. Although this finding was confirmed above all for lung cancer, it concerns all cancer types. The combination of age and bone metastases identifies a particular class of fragile patients. These patients should be given greater attention by planning closer evaluations, also through telehealth strategies.

Male and female young patients (≤55 years old) with bone metastases were evaluated for their risk of needing multiple remote visits. The RF model showed that male patients had an 11 times higher risk to need multiple remote consultations for pain management than female patients, especially for lung and colorectal cancers. These data must be interpreted very carefully and not only based on the possible gender/age differences in pain perception [27]. This is probably due to the low sample size. The ongoing creation of a larger dataset will allow us to carry out multivariate analyses and define whether other variables such as the differences in the type and stage of the tumor, the impact of the disease on functionality, psychosocial aspects, any comorbidities, as well as the associated anticancer therapies, may influence the data. Based on the results of this predictive analysis, we will develop more accurate pathways for addressing the multiple management problems of cancer pain. The proposed model of care, in fact, also provides for a multidisciplinary approach with the simultaneous involvement of different professionals, such as psychologists, physical therapists, surgeons, general practitioners, etc.

This study has several limitations. The small size of the dataset is the most important limitation. With a higher sample size, some variables could be more representative within the predictive processes. Nevertheless, since not many predictors were present in the models, analyses could be performed despite the limited number of patients.

Numerical variables contain more information than their categorial transformations. For example, the ROO variable could be used as discrete. For these reasons, we chose to categorize the cancer patients’ age into three categories (younger, middle-aged, and elderly patients). Another important limitation concerns the number of consultations per patient who is squeezed into the unit. Therefore, with a different distribution and the variables kept as numerical, better results would have been obtained.

The variables we adopted reflect our clinical practice. For example, although several opioids can be used for BTcP management [28], we usually prefer ROOs as the formulations licensed for this aim [29]. Moreover, this set of variables is not exhaustive. In cancer patients, for example, pain may not come from bone metastasis but often derives from invasion or abdominal metastasis, such as the peritoneal metastasis of colon cancer, and other causes. Probably, other variables such as cancer stage (e.g., TNM classification) should have been considered. In this regard, an improved dataset is planned in terms of its sample size and features. This will allow us to implement more sophisticated algorithms. The data from this study, however, can be useful for providing guidance for research in a field (telemedicine for cancer pain management) that is yet to be fully explored. The intervals between remote consultations, the needs for physical examinations, and the approaches for disease progression, as well as a careful definition of the process of early and simultaneous palliative care, are just some of the problems to be faced.

The methodological approach used in the simulations (i.e., SORs) has important potential that the clinician can exploit in predicting the outcome. On the other hand, SORs and simulations are obviously penalized by the sample size. In our analysis, the comparison between genders is an example of this gap. We highlight that their interpretation makes a practical and clinically useful sense if they are assessed through a good classifier.

Finally, we carried out only a few simulations as an example of the application of the evaluated model. The model, indeed, can be applied to a very large series of combinations of variables. Consequently, upon request, the dataset and model are available for further investigation.

## 5. Conclusions

The application of ML in telemedicine for pain management can enable physicians to make effective, real-time, and data-driven choices. This approach can be a key component in generating a better patient experience and improving health outcomes. A methodological approach to predictive analysis has great potential and could allow clinicians to provide important information to predict the outcome. Despite the important limitations of this study, in our analysis, the outcome (the number of remote consultations) was influenced by the selected variables such as the patient’s age, the cancer type, and the occurrence of bone metastases. Further studies are needed to design and refine this model of care for cancer pain patients.

## Figures and Tables

**Figure 1 jcm-11-05484-f001:**
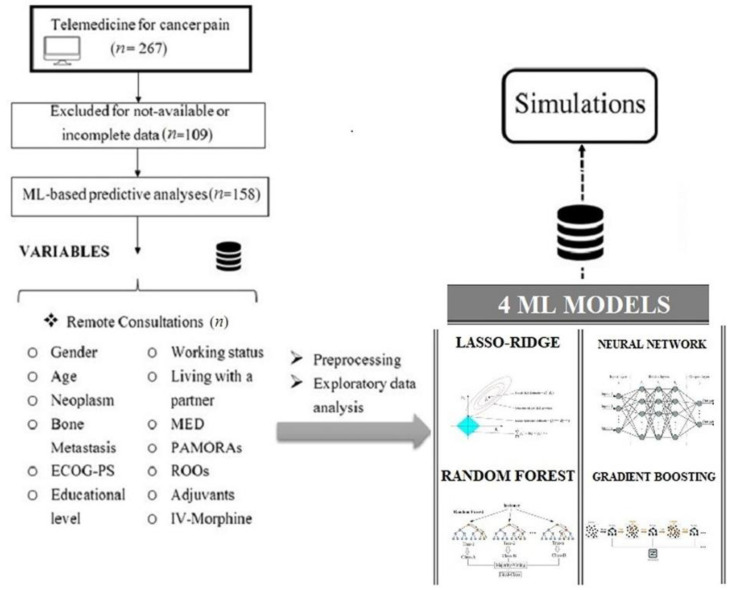
Flowchart of the study. Abbreviations: ML, machine learning; ECOG-PS, Eastern Cooperative Oncology Group Performance Status; MED, morphine-equivalent dose; PAMORAs, peripherally acting μ-opioid receptor antagonists; ROOs, rapid-onset opioids; IV-Morphine, intravenous morphine. Legend: the category “living with a partner” includes cohabitation and marriage.

**Figure 2 jcm-11-05484-f002:**
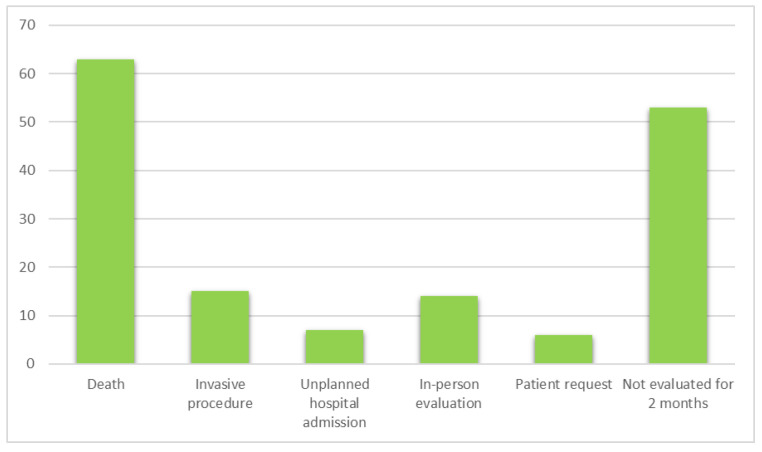
Reasons for interruption of the telemedicine pathway (*n* = 158).

**Figure 3 jcm-11-05484-f003:**
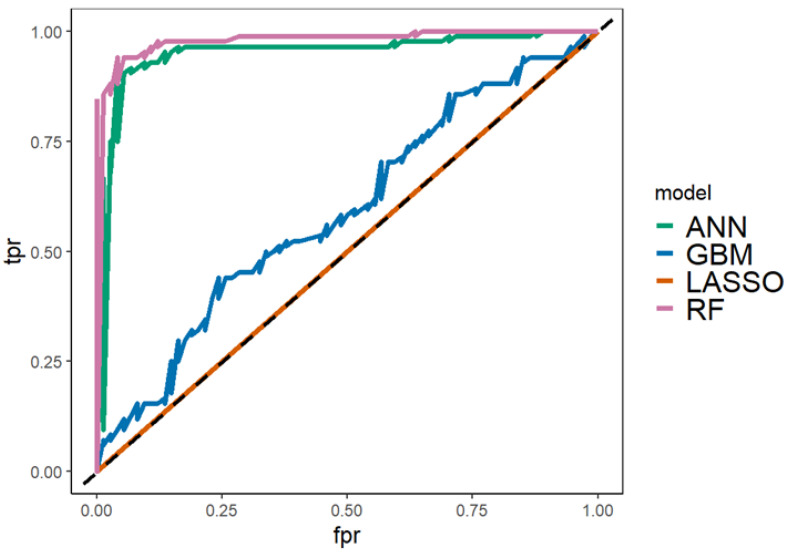
The area under the receiver operating characteristic (ROC) curve (AUC) of the considered models. False-positive rate (fpr) and true-negative rate (tpr) were considered. The plot shows the ROC curves calculated for each classifier over the entire dataset. RF and NN offer the best performance. Abbreviations: LASSO, LASSO–RIDGE regression; GBM, gradient boosting machine; ANN, artificial neural network; RF, random forest.

**Figure 4 jcm-11-05484-f004:**
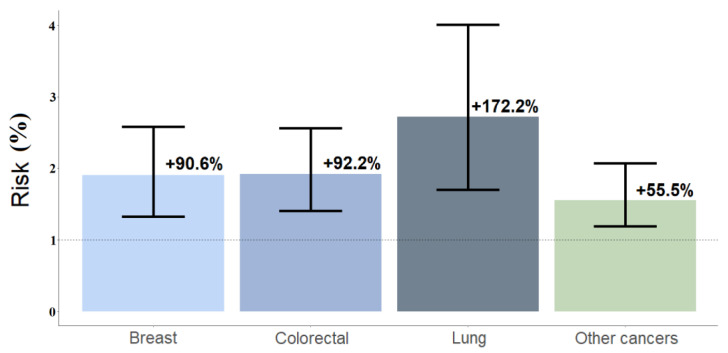
Simulation 1 refers to simulated odds ratios (SORs); percentages are labeled. This was performed for young patients (≤55 years old) with bone metastases and ROO use and young patients with bone metastases vs. those with no bone metastases. SORs for lung cancer were 2.72 (95%CI = 1.70–4.01); colorectal cancer 1.92 (95%CI = 1.40–2.56); other cancers 1.55 (95%CI = 1.19–2.07); breast cancer 1.91 (95%CI = 1.32, 2.58).

**Figure 5 jcm-11-05484-f005:**
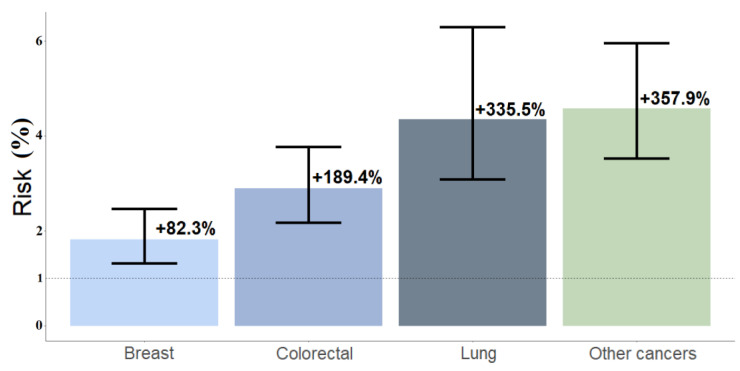
Simulation 2 refers to simulated odds ratios (SORs); percentages are labeled. This was performed for older patients (>75 years old) with bone metastases vs. patients without bone metastases. SORs for lung cancer were 4.35 (95%CI = 3.90–6.30); colorectal cancer 2.89 (95%CI = 2.18–3.77); other cancers 4.58 (95%CI = 3.53–5.95); breast cancer 1.82 (95%CI = 1.32, 2.47).

**Figure 6 jcm-11-05484-f006:**
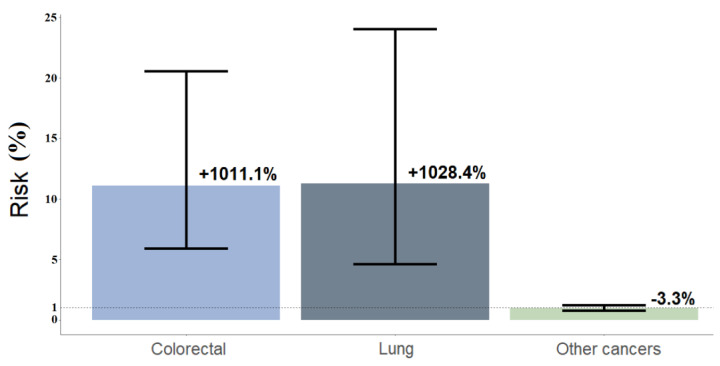
Simulation 3 refers to simulated odds ratios (SORs); percentages are labeled. It was performed for young individuals (≤55 years old) with bone metastases: male vs. female SORs. Young male patients had a significantly higher risk to receive multiple remote consultations when affected by lung cancer (SOR = 11.30, 95%CI = 4.60, 24.10) and colorectal cancer (SOR = 11.1, 95%CI = 5.90, 20.60). No statistical significance was found for other cancers (SOR = 0.97, 95%CI = 0.76, 1.23).

**Figure 7 jcm-11-05484-f007:**
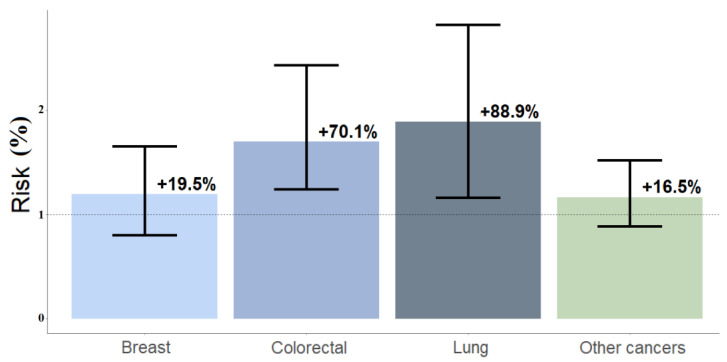
Simulation 4 refers to simulated odds ratios (SORs); percentages are labeled. This was performed for younger vs. older patients with bone metastases. Young patients had a significantly higher risk to receive multiple remote consultations for lung cancer. SORs were 1.89, 95%CI = 1.16, 2.82 for lung cancer and 1.70, 95%CI = 1.24, 2.43 for colorectal cancer. No statistical significance was found for other cancers (SOR = 1.16, 95%CI = 0.91, 1.52) and breast cancer (SOR = 1.19, 95%CI = 0.80, 1.65).

**Table 1 jcm-11-05484-t001:** Data collection and variables.

Data Collected	Variable(s)
Demographic and Social Information	Age Gender
	Working status (Y/N) Living with a partner * (Y/N) Education level
Clinical Data	Type of primary tumor Bone metastases ECOG-PS
Pain Therapy	MED Drugs for NP ROOs PAMORAs IV-Morphine
Remote Visits	Number

Abbreviations: ECOG-PS, Eastern Cooperative Oncology Group Performance Status; MED, morphine-equivalent dose; NP, neuropathic pain; ROOs, rapid-onset opioids; PAMORAs, peripherally acting μ-opioid receptor antagonists; IV-Morphine, intravenous morphine. Legend: * including cohabitation and marriage.

**Table 2 jcm-11-05484-t002:** Data from the considered variables.

Variable	*n* = 158 *
**Age** (years)	
Mean (SD)	63 (13)
Median (IQR)	65 (55, 72)
**Class of Age** (years old)	
≤55	43 (27%)
56–75	86 (54%)
>75	29 (18%)
**Gender**	
Female	81 (51%)
Male	77 (49%)
**Working Status** (*n* = 153)	
Not Working	110 (72%)
Working	43 (28%)
**Education Level** (*n* = 146)	
Secondary School	41 (28%)
High School	68 (47%)
Bachelor’s or Higher Degrees	37 (25%)
**Living with a Partner** (*n* = 153)	
Yes	107 (70%)
No	46 (30%)
**Neoplasm**	
Lung	22 (14%)
Colorectal	39 (25%)
Breast	21 (13%)
Others	76 (48%)
**Bone metastases** (*n* = 156)	
No	72 (46%)
Yes	84 (54%)
**ECOG-PS**	
ECOG-PS <3	84 (53%)
ECOG-PS = 3	74 (47%)
**MED**	
≤60 mg	64 (41%)
>60 mg	94 (59%)
**Assuming ROOs**	
No	114 (72%)
Yes	44 (28%)
**Assuming PAMORAs**	
No	125 (79%)
Yes	33 (21%)
**Assuming drugs for NP**	
No	78 (49%)
Yes	80 (51%)
**Assuming IV-morphine**	
No	146 (92%)
Yes	12 (7.6%)
**Remote consultations** (*n* = 158)	
Mean (SD)	2.27 (2.05)
Median (IQR)	2 (1, 3)
Min–Max for Patient	1–16
**Remote consultations (categories)**	
1	74 (47%)
>1	84 (53%)

Abbreviations: * *n* (%); ECOG-PS, Eastern Cooperative Oncology Group Performance Status; MED, morphine-equivalent dose; ROOs, rapid-onset opioids; PAMORAs, peripherally acting μ-opioid receptor antagonists; NP, neuropathic pain; IV-Morphine, intravenous morphine.

**Table 3 jcm-11-05484-t003:** Univariate analysis for data exploration.

	Remote Consultations		
Variable	one, *n* = 74 *	≥2, *n* = 84 *	*p*-value ^
Age (years)			0.019
*n*	74	84	
Mean (SD)	65 (13)	61 (13)	
Median (IQR)	68 (57, 75)	62 (53, 70)	
Class of Age (years old)			0.030
≤55	13 (18%)	30 (36%)	
56–75	44 (59%)	42 (50%)	
>75	17 (23%)	12 (14%)	
Gender			0.537
Female	36 (49%)	45 (54%)	
Male	38 (51%)	39 (46%)	
Working Status			0.987
No	51 (72%)	59 (72%)	
Yes	20 (28%)	23 (28%)	
(Missing)	3	2	
Education Level			0.374
Secondary School	22 (33%)	19 (24%)	
High School	31 (46%)	37 (47%)	
Graduation	14 (21%)	23 (29%)	
(Missing)	7	5	
Cohabiting/Marriage			0.711
Yes	50 (71%)	57 (69%)	
No	20 (29%)	26 (31%)	
(Missing)	4	1	
Cancer Type			0.516
Lung	8 (11%)	14 (17%)	
Colorectal	19 (26%)	20 (24%)	
Breast	8 (11%)	13 (15%)	
Others	39 (53%)	37 (44%)	
Bone Metastases			0.458
No	36 (49%)	36 (43%)	
Yes	37 (51%)	47 (57%)	
(Missing)	1	1	
ECOG-PS			0.396
<3	42 (57%)	42 (50%)	
=3	32 (43%)	42 (50%)	
MED			
<60 mg	33 (45%)	31 (37%)	
>60 mg	41 (55%)	53 (63%)	
Assuming ROOs			0.829
No	54 (73%)	60 (71%)	
Yes	20 (27%)	24 (29%)	
Assuming PAMORA			0.831
No	58 (78%)	67 (80%)	
Yes	16 (22%)	17 (20%)	
Assuming anti-NP Drugs			0.269
No	40 (54%)	38 (45%)	
Yes	34 (46%)	46 (55%)	
Assuming IV-Morphine			0.115
No	71 (96%)	75 (89%)	
Yes	3 (4.1%)	9 (11%)	

Legend: * *n* (%); ^ Wilcoxon rank-sum test; Pearson’s chi-squared test; significance at 95%. Abbreviations: ECOG-PS, Eastern Cooperative Oncology Group Performance Status; MED, morphine-equivalent dose; ROOs, rapid-onset opioids; PAMORAs, peripherally acting μ-opioid receptor antagonists; NP, neuropathic pain; IV-Morphine, intravenous morphine.

**Table 4 jcm-11-05484-t004:** Performance comparison of the different classifiers for the developed machine learning models.

Classifier	AUC	ACC (tr)	ACC (tst)	L	U	*p*	Sens (tst)	Spec (tst)	F1 Score	MCC
GBM	0.59	0.58	0.5	0.31	0.69	0.71	0.69	0.29	0.59	−0.03
RF	0.98	1	0.7	0.51	0.85	0.05	0.69	0.71	0.71	0.40
LASSO	0.5	0.53	0.53	0.34	0.72	0.57	1	0	0.7	-
ANN	0.95	1	0.57	0.37	0.75	0.43	0.5	0.64	0.55	0.14

Abbreviations: GBM, gradient boosting machine; RF, random forest; LASSO, LASSO–RIDGE regression; ANN, artificial neural network; AUC, area under the receiver operating characteristic curve; ACC (tr), accuracy on training; ACC (tst), accuracy on test set; L and U, 95%CI lower and upper limits of test set accuracy statistic; *p*, accuracy on the test set and relative test for significance; Sens (tst), sensibility on the test; Spec (tst), specificity on the test. MCC, Mathew’s Correlation Coefficient.

**Table 5 jcm-11-05484-t005:** Comparison (confusion matrix) for the two best models.

	RF		ANN	
	One	≥2	One	≥2
One	10	5	9	8
≥ 2	4	11	5	8

Legend: one or more consultations were considered. Abbreviations: RF, random forest; ANN, artificial neural network.

## Data Availability

All data were reported on a prospectively filled database and then registered on Zenodo [13].

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
