# Peer review of "Different Machine Learning Approaches for Implementing Telehealth-Based Cancer Pain Management Strategies"

_jcm, 2022, doi:10.3390/jcm11185484_

Round 1
Reviewer 1 Report
This is really exciting work, and I really enjoyed reading it.
Here are some of my suggestions
1. Introduction L52-55. Please use this paragraph to justify the reason for undertaking this work. What need would be fulfilled by predicting the needs of telemedicine consultation? 1, increasing access 2. On-demand services, 3. Plan for higher needs of patients.
2. Table 1. Please explain civil status,
In some places, you have used working status and job status interchangeably. Please be consistent with terminology.
You may also need to format the tables to improve readability.
3. Methods : 2.3.2: Machine learning models: Please include why you chose these four models and also references for this model.
4. Table 3. for univariable analysis for remote consultations: the third column just says more. Please consider making it >/= 2
Column one under education : graduation: please consider using a term that explains it better.
5. Discussion: please consider rewriting this section. For example, lines 274-275
Nevertheless, despite the growing use of telehealth methods, scientific evidence is still scarce to establish the correct pathways to be followed.
similarly, the sentence in line 278: is it an adapted care model vs Adopted? How is this model adapted
sentence is a line 281
I think one way to think about this would be how you think would be to talk about ML approaches and models and how to include these in your telemedicine practice. Finally, how should ML analysis inform telemedicine models considering higher needs in certain populations in their design?
Author Response
Reviewer 1
This is really exciting work, and I really enjoyed reading it.
Here are some of my suggestions
- ntroduction L52-55. Please use this paragraph to justify the reason for undertaking this work. What need would be fulfilled by predicting the needs of telemedicine consultation? 1, increasing access 2. On-demand services, 3. Plan for higher needs of patients.
Response. Thank you for your comment. We have completely revised this section according to your suggestions. The following sentences:
“In the planning of care pathways, AI represents a precious help to improve hospital workflows, identifying the activities that require priority and providing an adequate service to the patient's needs.” and “In the context of precision medicine for cancer pain management, ML algorithms are implemented to better customize treatment strategies. Because pieces of evidence are needed to establish the most appropriate telemedicine pathways, recognition of patients who require a greater number of remote visits can stimulate the planning of ad hoc processes for managing multiple care needs and calibrating resource allocation” .
were added to better explain the aim of the study and potential applications of the predictive analyses.
- able 1. Please explain civil status,
Response: thank you. We apologize for the mistake. We considered the category “Living with a partner”: it includes “Cohabitation and marriage”. Please, note that our aim was to evaluate the impact of telemedicine (n visits) in individuals with or without cohabitation with a partner. This data could be important for the management of the telemedicine process.
In some places, you have used working status and job status interchangeably. Please be consistent with terminology.
Response: thank you. We apologize for the mistake and used the “working status” for consistency.
You may also need to format the tables to improve readability.
Response: Although we improved their readability, the editing from JCM will certainly improve it.
- Methods : 2.3.2: Machine learning models: Please include why you chose these four models and also references for this model.
Response. We chose these four methods to compare different numerical approaches: LASSO-RIDGE is a binary regression model; RF and GBM are bagging and boosting methods for additive regression models; ANN is a function of functions algorithm which is now largely used for classification (and regression) problems, and we inserted it to evaluate if it worked better than others. Since the methods use different algorithms, our goal was to find the most useful to predict the output variable (i.e., number of remote consultations, as categorical).
Appropriate references were added.
- Table 3. for univariable analysis for remote consultations: the third column just says more. Please consider making it >/= 2
Response: we revised it.
Column one under education : graduation: please consider using a term that explains it better.
Response: we explained as “Secondary School”, “High School”, and “Bachelor’s or higher degrees”
- Discussion: please consider rewriting this section. For example, lines 274-275
Nevertheless, despite the growing use of telehealth methods, scientific evidence is still scarce to establish the correct pathways to be followed.
similarly, the sentence in line 278: is it an adapted care model vs Adopted? How is this model adapted sentence is a line 281
I think one way to think about this would be how you think would be to talk about ML approaches and models and how to include these in your telemedicine practice. Finally, how should ML analysis inform telemedicine models considering higher needs in certain populations in their design?
Response: Thank you a lot. By following your suggestions (and those from the other reviewers) we completely revised the DISCUSSION. The strategy was better addressed, as well as the aims of this investigation, and its potential applications (translational perspectives of the findings into clinical practice).
Reviewer 2 Report
Author aims to investigate the predictors of the number of remote consultations for cancer patients. The paper covers an important topic, relevant to the readers community. However, I have serious concern about the methodology and findings.
First, the outcome variable is a count variable however results represented as binary outcome. It should be clarified how the outcome categorized.
Second, 158 patients is a pretty small number to be generalized to whole cancer population and moreover to effectively use such ML methods. I also wonder why authors did not use simple logistic regression.
Third, the model evaluation metrics shows that none of the models had significant results accept RF. However the sensitivity found was lower than .70. Therefore I am skeptical that these models support author's conclusion "great potential and could allow clinicians to provide important information to predict the outcome".
Still, I think that authors have a meaningful motivation. I strongly suggest to increase the sample size and try logistic regression with more feature engineering.
Author Response
Reviewer 2
Author aims to investigate the predictors of the number of remote consultations for cancer patients. The paper covers an important topic, relevant to the readers community. However, I have serious concern about the methodology and findings.
First, the outcome variable is a count variable however results represented as binary outcome. It should be clarified how the outcome categorized.
Response: Thank you for your comment. Please note that our output variable (number of remote consultations) was originally given as count (numeric). Nevertheless, we observed that a lot of cancer patients had just one consultation, so a big left tail-distributed variable was drawn, and we assumed that analysis results could be useless. In particular, our aim was to evaluate the typology of cancer patients who require more than one remote consultation, so we decided to categorize it as one remote consultation, or more than one remote consultation.
Please, see the revised version of the DISCUSSION and how we addressed your highlighted concern.
Second, 158 patients is a pretty small number to be generalized to whole cancer population and moreover to effectively use such ML methods. I also wonder why authors did not use simple logistic regression.
Response: we are aware of this study limitation. Nevertheless, as you can note not many predictors were present in the models. Thus, analyses can be done despite the limited number of patients.
Logistic regression was almost included in lasso-ridge regression (lasso-ridge reducing the strength of betas).
Third, the model evaluation metrics shows that none of the models had significant results accept RF. However the sensitivity found was lower than .70. Therefore I am skeptical that these models support author's conclusion "great potential and could allow clinicians to provide important information to predict the outcome".
Still, I think that authors have a meaningful motivation. I strongly suggest to increase the sample size and try logistic regression with more feature engineering.
Response: RF approach was significantly the best one with respect to others (p=0.05), as we observed. The sensitivity of a binary classificator is its capacity to find a real case. We really expect great potential from that one if we will perform it with a larger dataset. The dataset nature (principally given by clinical features) is very hard to descry, and such a result is for us a good indicator of potential. We don’t forget that another (larger?) sample can surely change results and so our conclusion, but at this time RF largely represents the only classificator that could do that significantly not at random.
We are aware that sample size is maybe the strongest limitation of the study. As we wrote in the paper, our future purpose is to collect more data, also to keep the outcome variable as a count variable. The purpose of this publication was to begin stimulating research to structure telemedicine-based care pathways in this clinical setting.
Author Response
Reviewer 3
Overview
The piece of work written by Cascella and colleagues reports the use of validated machine learning models to predict the best course of action to treat cancer patients’ pain using telemedicine. Data has been collected from 158 patients, different machine learning algorithms were used, and random forest was the best among them for both training and test set. The authors have analysed 13 different variables and identified the most important among them to predict the number of remote consultations and concluded that this approach can be useful for clinicians in this context.
Please, find below my comments, questions and suggestions.
Introduction
The introduction section is short and sharp, clearly stating the purpose of the work in the end. However, more theoretical background could be added to shed more light onto the results presented and discussed by the authors.
Response: We appreciate the reviewer's comments. They express a high level of competence on the subject. The text has been completely revised and certainly, its value has improved a lot. Thanks again.
Starting from the first wave of the Covid-19 pandemic, for addressing cancer pain management, we structured a telemedicine-based model of care. This “hybrid” model is still used and provides for scheduled remote visits, but the patient can require other consultations. Additionally, in-person access is provided for emergencies or for diagnostic or clinical aims. Thus, for each patient, the number of telemedicine visits can vary depending on an unspecified number of reasons and we have noticed that some patients required a greater number of remote consultations.
Please note that we have included several sentences in the introduction and discussion sections to emphasize the purpose of the study. For example:
“… In the planning of care pathways, AI represents a precious help to improve hospital workflows, identifying the activities that require priority and providing an adequate service to the patient's needs.”…” In the context of precision medicine for cancer pain management, ML algorithms are implemented to better customize treatment strategies. Because pieces of evidence are needed to establish the most appropriate telemedicine pathways, recognition of patients who require a greater number of remote visits can stimulate the planning of ad hoc processes for managing multiple care needs and calibrating resource allocation”.
Furthermore, the structure of the methods and the presentation of the results (including the higher resolution of the figures) have been extensively revised. Several references were also added.
Material and methods
Section 2.3 – Predictive analysis. Lines 85-87 – this first sentence states that after pre-processing the data set, “models” were selected. Question: what do the authors mean by model selection right after the pre-processing?
Response: We apologize for the error. We corrected the word “models” as “variable”.
Please, make sure the original papers for each ML algorithm are properly cited.
Response: We added the references as required.
Section 2.3.2 – Please, consider changing the title of this section to “machine learning algorithms”, as “model” is what is obtained after using an algorithm that will generalize the data, will be assessed for goodness-of-fit, then tested against a test set, assessed again, then finally will be ready (or not) to be used to make predictions against new and unseen/unknown data.
Response: Thank you. We corrected it.
Line 92 – “Four ML-based prediction models”: see previous comment; the algorithms are either supervised or unsupervised, can be used to classification or regression purposes. The algorithm is a method, the model is the product of the algorithm, and the prediction is an “ability” of the model. Please, consider replacing “prediction models” by “algorithms”.
Response: Thank you. We corrected it.
Line 98 – replace “algorithms” by “algorithm”.
Response: Thank you. We corrected it.
Lines 107-110 - As the authors mentioned that ANNs needs a huge training set to find optimal weights:
Question: what was the method used by the authors to pick these algorithms and why did they use ANNs considering the small data set they have collected for this work?
Response: We know such a hurdle about our dataset. That’s why we chose to implement a single-layer ANN which is surely harder to train with respect to other methods.
We used four different ML methods for the same binary classification problem. We agree that other approaches can be implemented but we considered them as important and connected with a big mole of theory and statistical practical approaches.
Section 2.3.2
Please, describe the hyperparameters search methodology with more details, as well as the validation process.
Response: The methodology for hyperparameters is to be connected to empirical knowledge. However, we tried to include a very large set of them to find the best combination for each ML approach. The validation process is a repeated k-fold cross-validation method. We included specifications in the manuscript: “…each combination of hyperparameter was inserted as input for the algorithm. An 8-fold 5-repeated cross-validation method was adopted to find the best one, so the dataset was divided into 8 parts (20 individuals for any time), and training and test parts were performed for each combination and for 5 times; misclassification error rate was calculated upon 5 attempts (for a more precise managing of results)”
Please, include the formula used to calculate F1 score.
Response: we added it.
Please, add to the next version of the paper the calculated Mathew’s Correlation Coefficients (MCC) for each model to have a broaden view on their predictive power and robustness.
Response: we included it in the text (Methods) and in Table 4.
Please, consider adding the confusion matrix for each model, for both training and test sets, as it would add value to the comparison between the two best models.
Response: we just added the confusion matrix for the two best models because they had a perfect classification during the training phase. Another table was included (Table 5)
Results
This section must be reorganized according to the methods section. For example, the subsection 3.1 should be “data collection” or something related; instead, section 3.1 appears after all the data set description and is named “predictive analysis”.
Response: Thank you. We divided descriptive analysis (3.1) from predictive analysis (3.2).
There is a table containing results of a univariate analysis that is not described in the methods section. Please, describe it properly in the methods section.
Response: In the methods section, we reported: that “univariate analysis was performed to detect main associations of selected features with the outcome variable (remote consultations: one or more)”.
Table 4 list the models as “classifiers”.
Question: What are the classes the authors are trying to classify? Please, describe them in a clear way throughout the paper.
Response: At the beginning of the subsection “Model Processing and Evaluation”: “Since the predictive analysis was performed in order to predict which cancer patient should need to have more than one remote consultation, the outcome variable was “number of remote consultations” as dichotomized.”
Please, improve the quality of all figures in the results section, for better visualization of each graphic.
Response: We implemented their quality as required.
Section 3.2 – Simulations
It is not clear what the authors are willing to address with the paper in general. In the beginning of the article,
it has been defined as an objective to predict the number of the remote consultations for pain management of patients treating cancer. Please, consider changing the terminology for “risk analysis” in the very beginning of this section (lines 210-211). Further, consider moving the description of each simulation to the methods section and leaving the results of them in the results section.
Response: thank you for this evaluated suggestion. In the methods, we added a dedicated subsection (2.3.4 Risk analysis). Consequently, methods and results were revised regarding this issue.
Question: Is the increase in the number of remote consultations a risk?
Response: Although risk usually is given as a possibly dangerous condition, we think that it can be statistically observed as a propension to obtain a measurable quality. We just wanted to analyze the possibility to have more consultations as a fact, not positive or negative. Thus, we can call this section “risk analysis” as you considered.
Question: How did the authors predicted the data using their best model?
Response: As previously indicated, we revised the methods and results. Subsection 2.3.4 “Risk analysis” after “Model processing and evaluation” was included to better clarify our methodology.
Line 289 – EDA – please, explain the acronym (exploratory data analysis, I suppose)
Response: Corrected.
Discussion
Please, consider removing the subsection “limitations” heading but keeping the discussion about the limitations of the study.
Response. Corrected.
The entire section is missing some key points, for instance the impact of the increased number of remote consultations for cancer pain management over the public health programs or private healthcare/insurance; the way patients diagnosed with cancer deal with remote consultations, assessing the user-friendliness of the utilised platform for this end.
After reading the discussion, a question popped up: why is it important to predict the number of remote consultations for new patients? This could be also addressed in the discussion section.
Response: According to your comments and suggestions (and those of the other reviewers), this section has been completely rewritten. Other details were included in the introduction to clarify the model used and the prospects for implementing it. in this context, the purpose of the publication has been better specified.
Conclusion
The conclusion mentions “well-characterized variables” as a crucial point for the modelling. This could also be expanded by the authors in the discussion section, once it is part of the data quality and, as such, has a significant influence in the model’s predictive power, robustness and confidence.
Response: We assumed that numerical variables are “more powerful” than their relative categorization. We initially tried to perform algorithms with, for example, patients’ age as a numeric feature but sores appeared to shrink. Thus, the influence of the adopted variables was really significant for the power of predictions, and we discussed it. We hope we have explained everything about that in an exhaustive way. Moreover, we changed “well-characterized variables” to “selected variables”.
Reviewer 4 Report
Very nice and informative study
Materials and methods are well written
Just there are few questions in my mind
The aim of this study was to develop data-driven predictive models to identify those patients who may require more remote consultations. However, for cancer patients, remote consultations may not be suitable, even if there are advantages such as convenience and economy. Sometimes, pain indicates disease progression. Compared with in person evaluation, remote consultations lack of physical examination, which is a very important step for disease progression, such as ascites or intestinal obstruction of colon cancer causing by peritoneal metastasis leading to feeling abdominal distension, the liver metastasis of breast cancer, brain metastases of lung cancer leading to increased intracranial pressure. In these condition, intravenous morphine are not necessary while other medical intervention are needed. Whether telemedicine may cause delays should be considered.
In addition, as a supplement to in person evaluation, remote consultations should be conducted regularly, the intervals between remote consultations shoule be considered.
What’ more, TNM stage and clinical stage should also be considered in the model, at least in Tables 2 and 3. In many cases, the pain of cancer patients may not come from bone metastasis, but often from invasion or abdominal metastasis, such as peritoneal metastasis of colon cancer. Superior vena cava obstruction syndrome causes by breast cancer, and brain metastases from lung cancer.
Author Response
Reviewer 4
Very nice and informative study
Materials and methods are well written
Just there are few questions in my mind
The aim of this study was to develop data-driven predictive models to identify those patients who may require more remote consultations. However, for cancer patients, remote consultations may not be suitable, even if there are advantages such as convenience and economy. Sometimes, pain indicates disease progression. Compared with in person evaluation, remote consultations lack of physical examination, which is a very important step for disease progression, such as ascites or intestinal obstruction of colon cancer causing by peritoneal metastasis leading to feeling abdominal distension, the liver metastasis of breast cancer, brain metastases of lung cancer leading to increased intracranial pressure. In these condition, intravenous morphine are not necessary while other medical intervention are needed. Whether telemedicine may cause delays should be considered.
In addition, as a supplement to in person evaluation, remote consultations should be conducted regularly, the intervals between remote consultations shoule be considered.
What’ more, TNM stage and clinical stage should also be considered in the model, at least in Tables 2 and 3. In many cases, the pain of cancer patients may not come from bone metastasis, but often from invasion or abdominal metastasis, such as peritoneal metastasis of colon cancer. Superior vena cava obstruction syndrome causes by breast cancer, and brain metastases from lung cancer.
Response: We thank the reviewer for the valuable suggestions. Please note that the issues (for example, the appearance of bone metastases, or the need to include other variables such as the stage of the neoplasm) have been addressed in the different sections of the text and the manuscript has been completely revised. For example, we added the following sentences:
“... Moreover, the variables set is not exhaustive. In cancer patients, for example, pain may not come from bone metastasis but often derives from invasion or abdominal metastasis, such as peritoneal metastasis of colon cancer, and other causes. Probably, other variables such as cancer stage (e.g., TNM classification) should have been considered. In this regard, an enrichment of the dataset is planned in terms of sample size and features. This will allow us to implement more sophisticated algorithms. The data from this study, however, can be useful for providing guidance for research in a field (telemedicine for cancer pain management) that is yet to be fully explored. Intervals between remote consultations, needs for physical examinations, approaches for disease progression, as well as the careful definition of the process of early, and simultaneous palliative care are just some of the problems to be faced…”
Moreover, based on comments and concerns, other ameliorations were performed. We hope the reviewer will appreciate our effort to try to address such a broad topic from an original perspective and method (ML). Other research will have to clarify many aspects and help to structure safe and effective pathways.
Round 2
Reviewer 2 Report
I would like to thank authors for their thoughtful responses. I really like the motivation and goals of the paper. In order to investigate the methodology better I strongly suggest authors:
1. to follow the clinical ML reporting guidelines: an example can be found here: https://www.nature.com/articles/s41591-020-1041-y
2. Instead of p-values, reporting confidence intervals and make their conclusions based on them.
3. Double check their Figure 3 because the AUC in the figure looks much higher than the AUC reported in the table.
3. Describe the effects of each feature in the result. I suspect some features does not have any impact in the model and a model with only age might work the same.
Author Response
Thank you a lot.You played a key role in improving quality.
- Please consider that the six-point checklist proposed by Norgeot et al was followed as a guide for the description of the methods. However, some elements of the list, such as the definition of the clinical problem and technical steps, are described in other sections of the text. Furthermore, for descriptive convenience, other steps of the design were grouped and described together. For example, model processing and evaluation were addressed in a single subparagraph. Introducing the structure of the checklist at this stage would entail a complete revision of the text. Introducing the structure of the checklist at this stage would entail a complete revision of the text with an important departure from the original structure. We are grateful for the advice and we will follow this approach in the next article.
- Regarding the p values, we agree with the doubts, but the only one we show is the one in the table, because then the sORs are expressed as intervals.
- We carefully checked the table values and the result in the ROC.
- Since age as a numerical value obscures the other variables, we considered it appropriate to discretize. In other words, even if we lose information by discretizing, at least the other variables are all useful.
Reviewer 3 Report
Dear authors,
Thanks for addressing the points I have raised.
The formula for calculating MCC is missing and need to be added.
The quality of the graphics still need to be improved, specially the ROC curves.
Author Response
Thank you a lot.
We included the formula of MCC and improved the quality of the images.